# Evaluation of the Effect of a Recent Comparative Intradermal Tuberculin Test on the Humoral Diagnosis of Paratuberculosis Using Serum and Milk Samples from Goats

**DOI:** 10.3390/vetsci11030105

**Published:** 2024-02-28

**Authors:** Carlos Velasco, Javier Ortega, Alberto Gómez-Buendía, Anna Grau, Marisol López, Julio Álvarez, Beatriz Romero, Lucía de Juan, Javier Bezos

**Affiliations:** 1VISAVET Health Surveillance Centre, Complutense University of Madrid, 28040 Madrid, Spain; carvelas@ucm.es (C.V.); javior02@ucm.es (J.O.); agbuendia@ucm.es (A.G.-B.); jalvarez@ucm.es (J.Á.); bromerom@ucm.es (B.R.); dejuan@ucm.es (L.d.J.); 2Department of Animal Health, Faculty of Veterinary Medicine, Complutense University of Madrid, 28040 Madrid, Spain; 3Dirección General de Producción Agropecuaria e Infraestructuras Agrarias, Consejería de Agricultura y Ganadería de la Junta de Castilla y León, 47014 Valladolid, Spain; ana.grau@jcyl.es; 4Servicio Territorial de Agricultura, Ganadería y Desarrollo Rural, 05001 Avila, Spain; msol.lopez@jcyl.es

**Keywords:** caprine paratuberculosis, comparative intradermal tuberculin test, ELISA, booster effect, humoral diagnosis

## Abstract

**Simple Summary:**

This study addresses the suboptimal sensitivity of all the available ante mortem diagnostic techniques for paratuberculosis and evaluates the effect of a recent comparative intradermal tuberculin test on the humoral diagnosis of paratuberculosis in an infected caprine herd, as a potential diagnostic strategy to maximize the detection of infected animals. The results from our study showed an increase in the quantitative response detected using an ELISA technique (booster effect), which was translated into an increase, although not statistically significant, in the number of reactors.

**Abstract:**

Paratuberculosis (PTB) and tuberculosis (TB) are two mycobacterial diseases with a severe economic and health impact on domestic ruminants. The ante mortem diagnosis of PTB is hampered, among other factors, by the limited sensitivity of all the available diagnostic techniques. Since TB-infected goats subjected to the comparative intradermal tuberculin test (CITT) may experience a booster effect on their antibody titer and a potential enhancement to the sensitivity of humoral techniques for tuberculosis, in the present study we aimed to evaluate this diagnostic strategy on the humoral diagnosis of PTB in serum and milk samples collected from a caprine herd that was TB free and PTB infected. The results from 120 goats indicated a significant increase (*p* < 0.001) in the quantitative response detected using an ELISA technique, conducted using serum and milk samples taken 15 and 30 days after performing a CITT (day 0 of the study); although, it did not translate into a significant increase in the number of reactors during any of the testing events (0, 3,15, 30 and 60 days post-CITT). Additionally, the number of ELISA-positive animals was higher for the serum versus the milk samples at both 15 and 30 days post-CITT. The increase in the quantitative ELISA result suggested a diagnostic strategy that maximizes ELISA sensitivity, mainly using serum samples, in PTB-infected herds; although, it may depend on individual differences and the interpretation criteria.

## 1. Introduction

Mycobacterial infections, such as paratuberculosis (PTB) and tuberculosis (TB), are a major health problem in goat herds worldwide. Both diseases have a significant impact on animal health and economics, affecting animal welfare as well [1,2].

PTB, also known as Johne’s disease, is a chronic granulomatous enteritis caused by *Mycobacterium avium* subsp. *paratuberculosis* (MAP), and mainly affects domestic and wild ruminants [1]. However, the zoonotic risk of MAP and its relation to Crohn’s disease have also been suggested, causing inflammatory bowel disease (IBD) in humans [3,4]. PTB in goats results in chronic diarrhea, reduced dairy production, progressive weight loss, reproductive disorders and increased death rates on farms [1,5]. MAP infection mostly results from fecal–oral exposure at an early age and occurs through the ingestion of fecal material in contaminated colostrum, food, water or the environment [1,6], but also from transplacental transmission [7]. MAP-infected animals usually remain subclinically infected for years before the clinical phase and may shed the bacteria during this period, even prior to developing a detectable peripheral immune response, spreading the disease and making it very difficult to control [8]. However, goats are not usually submitted to PTB eradication programs in most countries including Spain and therefore PTB control is achieved, among others, through diagnosis and vaccination strategies against the disease, the latter being permitted and extended to caprine flocks in Spain [9,10,11].

Currently, ante mortem diagnostic methods for PTB in goats includes MAP detection through the polymerase chain reaction (PCR) [1,12,13], or bacteriological culture from feces, which has traditionally been considered “the gold standard” technique for MAP infection diagnosis [14]. However, this method has limitations, including its high cost [15] and the slow growth rate of MAP (of up to 6 months) [16]. Also, the sensitivity of fecal culture in clinical animals is 70%, but it is only 23–29% in subclinical animals [16]. This lower sensitivity is due, among other factors, to the intermittent fecal excretion of the bacteria [17,18]. Therefore, additional ancillary tests based on the detection of the humoral response, such as those based on the enzyme-linked immunosorbent assay (ELISA) technique, have been extensively applied for ante mortem diagnosis, due to the relatively low cost and quicker results than for culture [16]. Nevertheless, limitations in the sensitivity of the ELISA technique have been reported, particularly in the early stages of infection when animals do not develop a detectable humoral immune response, which may lead to false-negative reactions [15,16]. Furthermore, limitations in the specificity of the ELISA technique have also been reported [19], mainly due to the shared antigens between MAP and other mycobacteria, such as the causal agents of caprine TB [20,21]. In this context, previous studies have reported an increased risk of false-positive reactions using the ELISA technique in TB-infected animals, probably due to cross-reactions issues [22].

Caprine TB is a zoonotic disease caused by members of the *Mycobacterium tuberculosis* complex (MTBC), mainly *Mycobacterium bovis* and *M. caprae* [23]. Unlike TB in cattle, caprine herds are not submitted to national eradication programs within the EU, while in Spain, several regions have implemented specific eradication and surveillance programs [24]. These programs are primarily based on test and cull strategies. The former involves performing the single and comparative intradermal tuberculin test (SITT and CITT, respectively), periodically (Orden AYG/415/2016). The SITT is based on cervical or scapular intradermal inoculation with an *M. bovis* purified protein derivative (bovine PPD) and, simultaneously, with *M. avium* subsp. *avium* PPD (avian PPD) via a CITT, to assess the type IV hypersensitivity response elicited after 72 h at the PPD inoculation site [25]. Goats subjected to a CITT may experience a booster effect on their antibody titer, potentially enhancing the sensitivity of the serological test for the diagnosis of TB, based on the detection of specific antibodies in milk and serum samples, maximizing the detection of TB-infected animals [26]. In this sense, the objective of the present study was to evaluate, for the first time in goats, the effect of a recent CITT on the humoral MAP response in serum and milk samples, to assess its usefulness as a potential diagnostic strategy to maximize the detection of PTB-infected animals.

## 2. Materials and Methods

### 2.1. Study Design

For the study, female animals (aged 1 to 2 years) in lactation (calving throughout January 2023) were randomly selected from a historically TB-free dairy herd (n = 120) of Murciano–Granadina breed goats, located in Castilla y León (the central region of Spain). The herd tested negative in the annual tests performed in the last seven years, within the framework of a regional eradication program, and had no previous history of TB (the last test being on 5 September 2022). Also, the herd had never been subjected to a vaccination program against PTB and had a 10% apparent prevalence of PTB (based on a serum ELISA test), as a result of a previous routine screening performed by the official veterinary service in Castilla y León, 6 months before this study was performed (5 September 2022). Additionally, the presence of MAP in the herd was confirmed using a PCR for environmental DNA sampling using sponges, as described elsewhere [27]. Briefly, environmental samples (n = 5) were collected from the floor of the milking room, salt stone, bulk tank, trough and sprue. Later, MAP was detected in the total samples collected using a conventional PCR.

Firstly, the goats in the herd were subjected to a SITT and CITT at day 0 of the study (12 March 2023). Secondly, serum and milk samples were collected from the animals prior to the intradermal tests (day 0) and on days 3 (15 March), 15 (28 March), 30 (12 April) and 60 (12 May) post-CITT. The serum and milk samples were analyzed using a commercial indirect ELISA kit, in order to evaluate the Ab levels in this type of samples, throughout the days sampled.

The animals included in this study were not experimental animals. All the handling and sampling procedures were carried out by veterinary staff, in accordance with local and Spanish legislation (Royal Decree 2611/1996).

### 2.2. Serum and Milk Sample Collection

Blood samples were collected, through a jugular venipuncture, into tubes with no additives. Then, the blood samples were centrifuged (1500× *g* for 10 min), and serum was collected and stored at −20 °C, until the ELISA assay was performed. Milk samples (30 mL) were collected from the animals during daily milking. Afterwards, the milk samples were centrifuged (13,000× *g* for 5 min) and 1 mL of whey was collected and stored at −20 °C, until the ELISA testing.

### 2.3. Intradermal Tuberculin Test

A SITT and CITT were carried out according to Regulation EU 2016/429 and the standard operating procedures (SOPs) of the European Union Reference Laboratory for bovine TB for intradermal tuberculin testing in caprine animals (SOP/002/EURL; https://www.visavet.es/bovinetuberculosis/databases/protocols.php (accessed on 1 February 2024)). For the SITT, goats were intradermally inoculated with 0.1 mL of bovine PPD (CZ Vaccines, Porriño, Spain) on the left-hand side of the neck using a Dermojet syringe (Akra Dermojet, Pau, France), together with 0.1 mL of avian PPD, on the right side of the neck, for the CITT. In the case of the SITT, an animal was classified as positive if an increase in the skin fold thickness ≥4 mm and/or the presence of clinical signs, such as exudation, pain, oedema or necrosis, occurred. An animal was classified as positive for the CITT, when the bovine reaction was >4 mm greater than the avian reaction and/or there were clinical signs, like exudation, pain, oedema or necrosis, at the bovine PPD injection site.

### 2.4. Indirect ELISA

The serum and milk samples were analyzed and interpreted using an indirect commercial ELISA kit, ID Screen Paratuberculosis Indirect screening test (Innovative Diagnostics, Grabels, France), for the detection of anti-MAP IgG. The ELISA kit involves a primary absorption step, using an extract of *Mycobacterium phlei* to reduce the cross-reactivity with environmental mycobacteria, as described elsewhere [28]. The results were interpreted following the manufacturer’s recommendations; animals were considered positive if the sample-to-positive percentage result (S/P%) was greater than 70% and 30% for the serum and milk samples, respectively. The sensitivity and specificity attributed to the ELISA technique, in previous studies in goats [29], is 74.3% (95% CI: 59.8–88.8%) and 98.6% (95%CI: 96.6–100%), respectively, when using serum samples, and 60% (95% CI: 43.8–76.2%) and 99.3% (95% CI: 97.8–100%), respectively, when using milk samples.

### 2.5. Statistical Analysis

Comparisons between the number of goats reacting to the ELISA test in the different testing events and in both samples (serum and milk) were evaluated using Cochram’s Q test and McNemar’s test, respectively. The quantitative results (the S/P% values on different days) were compared using the Friedman test. All the statistical tests were carried out using the SPSS 27 commercial statistical software (IBM, New York, NY, USA). A *p* value of 0.05 was considered statistically significant. 

## 3. Results

None of the animals included in the study had reactors to the SITT or CITT. Regarding the ELISA results, sixteen (13.3%) and six (5.0%) out of 120 goats were positive in the ELISA at day 0 of the study, using the serum and milk samples, respectively. The highest prevalence rate of positive goats in the ELISA test was observed in the sampling conducted on day 15 post-CITT when using the serum (21/120 (17.5%)) and milk (10/120 (8.3%)) samples, although this increase was not statistically significant with regard to the initial day of the study (Table 1). 

Regarding the sample analyzed, the number of positive goats in the ELISA test was significantly higher (*p* < 0.005) for all the testing events when using serum compared to milk samples. With regard to the quantitative results (S/P%), no statistically significant differences were observed in the sampling conducted on day 3 post-CITT, regardless of the sample analyzed. Afterwards, a significantly (*p* < 0.001) higher increase in the S/P% values (booster effect) were observed in day 30 compared to day 0 when using the serum samples. An earlier booster effect was observed when the milk samples were used, with significant differences (*p* < 0.001) in the S/P% values on day 15, which were also maintained until day 30 post-CITT in comparison to the samples collected on day 0. After day 30, a progressive decrease in the S/P% outcomes were observed for both samples, up to similar values recorded on day 0 (Table 1).

For the subset of ELISA reactors on day 0 (Figure 1A), no significant differences in the S/P% values recorded for the different sampling events were observed. This finding could be related to the high values observed in these positive animals at day 0 (see Appendix A). However, in the group of ELISA-negative goats on day 0, significant differences (*p* < 0.001) were found in the S/P% values in the serum and milk samples collected on day 15 and 30 in comparison to day 0 (Figure 1B), which explains the finding observed for the total number of animals.

## 4. Discussion

In this study, we aimed to elucidate the impact of a recent CITT on the humoral response against MAP in serum and milk samples, collected at several different times, from a caprine herd that was PTB infected and TB free. The results from the present study showed an increase in the S/P% values in an ELISA test performed on the serum and milk samples collected 15 and 30 days after performing a CITT. These findings were attributed to an enhanced antibody response as consequence of PPD inoculation [26,30,31,32]. However, the increase in quantitative reactivity did not translate into a significant increase in the number of ELISA reactors, even when using stringent cut-off points not recommended in our country [33] (data not shown). Nevertheless, our results must be interpreted carefully due to the limited sample size, which provided only limited potency (~60%) allowing only large changes to be detected in infected animals using the ELISA test after the CITT (e.g., increases from 50 to 85% with a 15% error). Hence, changes in the number of reactors may occur in other species or epidemiological settings. In this sense, previous studies have reported a booster effect in antibody responses measured via milk and serum ELISAs in PTB-infected cattle herds [33,34]. Kennedy et al. [33] reported a significant increase in the number of cattle reacting to an ELISA test on days 14 and 9 post-PPD inoculation in serum and milk samples, respectively, indicating a faster increase in the responses in milk versus serum samples, similar to what was observed here. Also, in the study by Kennedy et al. [33], the number of positive cattle in the ELISA test when using serum samples with regard to the day of the CITT was significantly higher until day 58 post-CITT, compared to day 30 post-CITT observed in our study. This marked and long-lasting reactivity (the latter particularly in serum samples) could be associated with individual factors in the animals included in the study, such as different stages of infection.

A strong difference in the type of sample used was noted here, with a higher number of positive goats in serum versus milk samples in all testing events, thus suggesting a higher sensitivity when using this type of sample. Salgado et al. [29] also reported a higher sensitivity in the ELISA test when using serum samples compared with milk samples, using fecal culture as the reference test (74.3% (95% CI: 59.8–88.8%) and 60% (95% CI: 43.8–76.2%), respectively). The different sensitivity observed in serum and milk may be due to different IgG production in these types of samples or the IgG subclass targeted by the technique. In this sense, a previous study reported differences in IgG titers and subclasses in different types of samples, such as serum, milk or colostrum [35]. Also, previous studies have reported a higher antibody response at early and late lactation stages and, thus, a higher sensitivity of the ELISA test at this time [36,37]. In this context, considering that the goats in our study were not at the beginning or end of lactation, the effect of the lactation stage on the sensitivity of the ELISA test using milk samples cannot be discarded. 

The difficulty in finding goat herds in Spain that are either not infected with PTB or not vaccinated against the disease hinders the possibility to conduct detailed analyses on the impact of specificity. In this sense, the interference caused by a recent CITT in the humoral diagnosis of PTB has been established in cattle, leading to false-positive reactions using serum [19] and milk samples from PTB-free herds [34]. 

However, since PTB infection was confirmed in the goat herd analyzed here and it is widely known that the sensitivity of antibody-based diagnosis of this disease is limited [16,38], the increase in the number of reactors recorded after the CITT could have been due to an increased sensitivity rather to a decreased specificity. For this reason, additional studies in TB and PTB-free caprine herds (that have not been vaccinated against the disease) would be of paramount importance.

## 5. Conclusions

In conclusion, this study is the first to characterize the effect of a previous CITT on the humoral diagnosis of PTB under field conditions. We demonstrated an increase in the quantitative responses in the ELISA test using serum and milk samples collected 15 or 30 days after a CITT, which translated into an increase, although not statistically significant, in the number of reactors. Additional studies considering other epidemiological scenarios are desirable, to determine if this could be the basis of diagnostic strategies aimed at maximizing the detection of PTB-infected goats in the context of TB eradication programs based on intradermal testing.

## Figures and Tables

**Figure 1 vetsci-11-00105-f001:**
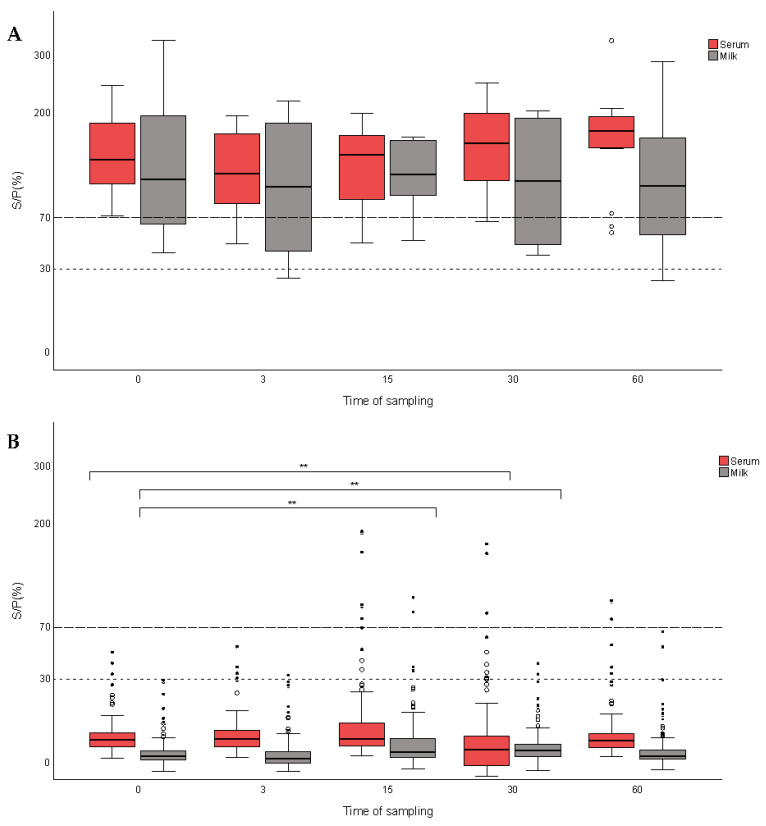
Quantitative ELISA results (S/P%) in serum (red) and milk (grey) at days 0, 15, 30 and 60 of the boosting effect study in positive (**A**) and negative (**B**) goats compared to the ELISA on the initial day. Upper striped and lower dotted lines represent the cut-off point of 70% and 30% S/P for serum and milk samples, respectively. Significant differences were obtained between the S/P% values on day 0 and 30 for serum sampling and between the S/P% values on day 15 and 30 with regard to the initial day for the milk samples. ** *p* < 0.001.

**Table 1 vetsci-11-00105-t001:** Number (n) and proportion of positive goats (%) in the ELISA test and statistical differences (*p* value) in the S/P% results at different times with regard to the initial day of the study (day 0).

		Number of Rectors in the ELISA (n)				Quantitative Response (S/P%)	
Group	Sampling Day	Serum		Milk		Serum	Milk
		n (%)	*p* Value	n (%)	*p* Value	*p* Value	*p* Value
Total (n = 120)	Day 0	16/120	-	6/120	-	-	-
	Day 3	15/120	1.0	6/120	1.0	1.0	0.37
	Day 15	21/120	0.225	10/120	0.35	0.66	<0.001
	Day 30	18/120	1.0	8/120	1.0	<0.001	<0.001
	Day 60	17/120	1.0	7/120	1.0	1.0	1.0

## Data Availability

The data presented in this study are available in the Appendix A.

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
