# Peer review of "Evaluation of the Effect of a Recent Comparative Intradermal Tuberculin Test on the Humoral Diagnosis of Paratuberculosis Using Serum and Milk Samples from Goats"

_vetsci, 2024, doi:10.3390/vetsci11030105_

Round 1
Reviewer 1 Report
Comments and Suggestions for Authors
Review vetsci-2853391-peer-review-v1
There are two main concerns with this paper. Firstly, the study population is too small to show the likely increase in seroprevalence observed after the TB test. Secondly the ethical approval to the study does not appear to have been adequate given the nature of the interventions. Both are discussed further below.
Title: No comment
Abstract:
Line 18: “This study addresses the need of….” The study does not enter into a comparison of existing diagnostic techniques and this text is therefore misleading.
Line 29: “from a TB-free and PTB-infected herd”. This is open to misinterpretation and would be better described as a herd that was TB-free and PTB-infected.
Line 35: The concluding sentence claims something that this limited study cannot demonstrate, because too few animals were involved in the study.
Introduction
In general the first section introduces the issue as it applies to cattle. There is no mention of how goats may differ from cattle. While the literature in goats is far smaller than cattle there still needs to be some explanation of the disease and diagnostics in goats.
Line 44: ref 1. This reference does not appear to be work done to assess the impact of PTB on welfare. Rather it examines the effects of vaccination on diagnostics.
Materials and Methods
Line 75: The description of the age of the females is confusing. Presumably it is adult animals of less than two years of age. What is adult in goats? Why are the older animals in the herd not included?
Line 76-77: The last time the TB test was carried out before the onset of this study and whether this was the routine surveillance test or an additional test; the date that the samples were taken to determine the seroprevalence in relation to the onset of the study; and the dates of the study are all required.
Line 80: Reference 16 is on sampling cattle and their environment. A description of how the goat herd and its environment was sampled should be given.
Line 85: These handling and sampling procedures are quite intensive and it is not clear whether the legislation covers the ethical issue of exposing the animals to unnecessary procedures for experimental reasons.
Results
Line 123: “The highest record…” is vague and it should be seroprevalence (serum) or prevalence of antibody positive animals (milk and serum).
Line 125: The increase in seroprevalence from day 0 to day 15 is 31% and is not small. The sample size of 120 is too small to show that such a difference would be significant. It would not be until the sample size was increased to several hundred that the observed difference in seroprevalence would be likely to be significant at the 5% level. For a sample size of 120 the increase in seroprevalence would have to approach 75% to achieve significance.
Discussion
Line 157: refs 15, 16, 17. It needs to be explained why these references are quoted as otherwise this is meaningless.
Line 172: “A strong effect of the sample used…”. This is not an effect, but simply an observed difference in the outcome (of the effect of the TB test) when measured by milk or serum.
Lines 178 and 179: Ref 22. This paper found no statistical difference between milk and serum antibody positives in a large study. While the difference in subclasses of IgG is mentioned, it does not explain why serum would be more likely to have a higher S/P% than milk. The Lombard article does focus on the impact of stage of lactation on the antibody concentration in milk, which may also be of relevance in goats and therefore for this study.
Lines 189-194: It concludes that this allows “discarding a potential strategy to increase the ELISA sensitivity". However the data are not extensive enough to allow this conclusion and the observed difference is not small. In a larger study if the same difference was observed then it would be likely to be significant. Further it should be noted that many lactating goat herds are significantly larger than the current study population and if this difference in seroprevalence was observed it would be significant and could lead to a useful increase in test sensitivity and a potential to improve control.
Institutional Review Board Statement
Lines 208-210: This is a confusing sentence. Does it mean that the board thought it unnecessary to approve the work? Does that mean that the ethical aspects of this study have not been properly considered? What is meant by opportunistic blood and milk samples? Was there another experiment going on at the same time involving these animals?These animals were required to be handled on a total of five occasions to be blood sampled. It is difficult to conceive this intervention as being required for anything other than the experiment.
*My concerns are that the need for internal ethical approval was waived because the collecting of samples was “opportunistic”.
Institutional Review Board Statement: Ethical review and approval were waived for this study due to opportunistic blood and milk samples taken from animals by veterinary staff and with the farmer´s and owner´s permission. 1. The study required the TB test to be carried out and follow up handling for blood samples on four occasions. It did not state whether the TB test was the scheduled test for statutory disease control, or one specifically carried out for the study. The blood samples were clearly carried out specifically for the study and were not carried out to benefit the 120 animals involved in the study. There is therefore a clear need for ethical approval and for clarification on whether the TB test was carried out specifically for the study. 2. The explanation that the ethical approval was waived because the collecting of samples was “opportunistic” is difficult to understand not least because this is an unusual use of the word opportunistic. In what way was the collection of the samples “opportunistic”. This particular reason may justify approving work, but to waive the need for ethical approval is not logical. 3. If the samples were “opportunistic” there is a clear implication that the animals were being handled for blood sampling for some other reason. This should be explained. If they were not being handled and blood sampled for some other reason these samples were not “opportunistic”.Comments on the Quality of English Language
Minor edits only.
Author Response
Reviewer 1:
There are two main concerns with this paper. Firstly, the study population is too small to show the likely increase in seroprevalence observed after the TB test. Secondly the ethical approval to the study does not appear to have been adequate given the nature of the interventions. Both are discussed further below.
Thank you very much for all your comments and suggestion to improve the quality of the manuscript. We have considered all of them. Our responses to your questions/comments are detailed below.
- Abstract:
Line 18: “This study addresses the need of….” The study does not enter into a comparison of existing diagnostic techniques and this text is therefore misleading.
Thank you. Changes has been included in order to gain clarity (present line 18-19).
Line 29: “from a TB-free and PTB-infected herd”. This is open to misinterpretation and would be better described as a herd that was TB-free and PTB-infected.
Corrected (present lines 30-31).
Line 35: The concluding sentence claims something that this limited study cannot demonstrate, because too few animals were involved in the study.
A previous analysis was conducted to determine the minimum number of animals necessary to reach conclusions (see below for explained simulation analysis in the sample size). Indeed, finding goat herds without tuberculosis but with paratuberculosis (and not vaccinated against the disease) and at a reasonable distance from the laboratory is very challenging, which is why the study focused on a single herd. However, it is true that the observed quantitative differences do not rule out the possibility that under different epidemiological conditions or using other diagnostic kits and interpretation criteria (e.g. in other countries), there may also be a qualitative effect. In this regard, we have made some modifications to the text to suggest this possibility (present lines 36-38; 213-214). Nevertheless, based on the results, we believe that performing an intradermal test for caprine tuberculosis to increase the sensitivity of the paratuberculosis antibody detection technique does not seem to be a clearly useful strategy, as has been demonstrated in previos studies, for example, in the humoral diagnosis of caprine tuberculosis (Ortega et al., 2022. Factors affecting the performance of P22 ELISA for the diagnosis of caprine tuberculosis in milk samples. Res Vet Sci 145; 10.1016/j.rvsc.2022.02.008).
- Introduction
In general the first section introduces the issue as it applies to cattle. There is no mention of how goats may differ from cattle. While the literature in goats is far smaller than cattle there still needs to be some explanation of the disease and diagnostics in goats.
Personally, we believe that the first version of the manuscript already had an introduction focused enough on goats; nevertheless, we have attempted to focus it even more on this species (present lines 48-62 and 81-84). There is less information available regarding goats because there have not been as many studies conducted in cattle. However, our research group has precisely aimed to increase knowledge about the diagnosis of mycobacterial diseases in goats. In this sense, studies such as this one or others before it have demonstrated that it is not always possible to extrapolate results regarding mycobacterial diagnostic techniques from cattle to goats.
Line 44: ref 1. This reference does not appear to be work done to assess the impact of PTB on welfare. Rather it examines the effects of vaccination on diagnostics.
Thank you for the suggestion. We have removed the reference and included two extensive reviews of tuberculosis and paratuberculosis in goats, as we believe it can fit better with the animal health, economic and animal welfare issues mentioned in the introduction (present lines 45-47).
- Materials and Methods
Line 75: The description of the age of the females is confusing. Presumably it is adult animals of less than two years of age. What is adult in goats? Why are the older animals in the herd not included?
Thank you. We have made some changes for clarity (present lines 100-101) . By 'adult,' we were referring to animals of sexual maturity, but we have removed the term and specified the age range. We aimed for a relatively homogeneous population, so lactating females between 1 and 2 years old were selected. They needed to produce milk and furthermore, intradermal tuberculin tests cannot be performed until they are older than 45 days. We did not use older animals because there were fewer of them on this particular farm, and there was greater disparity in ages. We also preferred to choose all animals from their first or second lactation.
Line 76-77: The last time the TB test was carried out before the onset of this study and whether this was the routine surveillance test or an additional test; the date that the samples were taken to determine the seroprevalence in relation to the onset of the study; and the dates of the study are all required.
All the analysis were made in the context of the compulsory/voluntary TB/PTB control programs in goats. The testing event referred by the reviewer was carried out 6 months before the study by official veterinary services. We have included additional data for better understanding (present lines 102-105 and 114-117).
Line 80: Reference 16 is on sampling cattle and their environment. A description of how the goat herd and its environment was sampled should be given.
The methodology of using sponges for environmental sampling, as well as the subsequent extraction of genetic material and PCR, is independent of the animal species and can be done similarly. Since this work is a brief report, we consider it is not necessary to detail it as it can be performed similarly to what has been described in that previous study, and there are no differences between species. In any case, we have included some relevant information (present lines 110-113) but if it is deemed indispensable to include the methodology, we can transcribe it into this manuscript in subsequent revisions.
Line 85: These handling and sampling procedures are quite intensive and it is not clear whether the legislation covers the ethical issue of exposing the animals to unnecessary procedures for experimental reasons.
Our research group has extensive experience in conducting experimental studies and clinical trials. In fact, among the authors are responsible for the animal experimentation and clinical trial services at the VISAVET research center, which is also the European Reference Laboratory (EU-RL) for bovine tuberculosis of the European Union and for tuberculosis in mammals of the WOAH. We have authored dozens of research manuscripts that have required approvals from ethics and animal experimentation committees, something we are deeply committed to and have always adhered to. In several similar studies, we consulted our regional experimentation committees and were advised that, due to the nature of the study, obtaining permission was not mandatory. In fact, we have recently published a few very similar studies in terms of methodology where such permission was also not required (Ortega et al., 2022. Factors affecting the performance of P22 ELISA for the diagnosis of caprine tuberculosis in milk samples. Res Vet Sci 145; doi: 10.1016/j.rvsc.2022.02.008) (Velasco et al., Effect of a recent intradermal test on the specificity of P22 ELISA for the diagnosis of caprine tuberculosis. Front Vet Sci 11; doi.org/10.3389/fvets.2024.1358413). Of course, we understand the reviewer's concern, but in our case, the final decision to request such permissions lies with our authorized organization and depend of its interpretation. Additionally, among the authors of the study are personnel from official veterinary services, which further underscores our commitment to complying with current legislation regarding purely experimental studies.
- Results
Line 123: “The highest record…” is vague and it should be seroprevalence (serum) or prevalence of antibody positive animals (milk and serum).
Corrected (present line 168).
Line 125: The increase in seroprevalence from day 0 to day 15 is 31% and is not small. The sample size of 120 is too small to show that such a difference would be significant. It would not be until the sample size was increased to several hundred that the observed difference in seroprevalence would be likely to be significant at the 5% level. For a sample size of 120 the increase in seroprevalence would have to approach 75% to achieve significance.
Thank you for your comment. We agree with the reviewer in the limitation of the study imposed by the low sample size. Through simulation we have estimated that even assuming an increase of 35% from a basal sensitivity of 50% and accepting a 15% error, our sample size would only detect the occurrence of these difference assuming independence between test results around 60% of the times. Still, our sample size is slightly above the minimum sample (n=79) size according to P. A. Lachenbruch (On the sample size for studies based upon McNemar's test. Stat Med. 1992, 11(11):1521-5. DOI:10.1002/sim.4780111110) for detecting significant differences with an 80% power and a 95% confidence level in a McNemar test for paired groups assuming a 10% increase in the probability of detecting an infected animal. Therefore, considering the difficulties in accessing a large sample of animals with a known TB and PTB status and at a reasonable distance from the laboratory, it hinders the possibility to conduct larger studies. Moreover, although in this study we were not required to request animal experimentation permission, for welfare reasons, we always employ the minimum number of animals necessary to confirm the hypothesis we are testing.
- Discussion
Line 157: refs 15, 16, 17. It needs to be explained why these references are quoted as otherwise this is meaningless.
We understand that you are referring to references 15, 17, and 18. These references are included because they are manuscripts that demonstrate the effect of a previous intradermal tuberculin test as a strategy to increase the sensitivity of Ab based tuberculosis diagnosis (booster effect). It is true that there are several studies justifying this aspect, and they could be consolidated into a single reference, thus avoiding any suspicion of self-citation, as these are studies conducted directly by us or in which we have been able to participate in some way. In addition, we have included additional studies evaluating the booster effect, including goats.
Line 172: “A strong effect of the sample used…”. This is not an effect, but simply an observed difference in the outcome (of the effect of the TB test) when measured by milk or serum.
Thank you. Corrected (present line 225)
Lines 178 and 179: Ref 22. This paper found no statistical difference between milk and serum antibody positives in a large study. While the difference in subclasses of IgG is mentioned, it does not explain why serum would be more likely to have a higher S/P% than milk. The Lombard article does focus on the impact of stage of lactation on the antibody concentration in milk, which may also be of relevance in goats and therefore for this study.
We wanted to underscore that the differences in the IgG titer or the IgG subclass targeted by the ELISA in serum and milk may be related with different S/P% values and definitelty, differences in serum and milk ELISA sensitivity. However, to clarify this idea we have included a recent comparative study evaluating differences in IgG production in different samples (present lines 232-233). Also, in line with your suggestion, we have included in the discussion a potential impact of the lactation stage on the antibody titer and therefore the sensitivity of the ELISA (present lines 233-237).
Lines 189-194: It concludes that this allows “discarding a potential strategy to increase the ELISA sensitivity". However the data are not extensive enough to allow this conclusion and the observed difference is not small. In a larger study if the same difference was observed then it would be likely to be significant. Further it should be noted that many lactating goat herds are significantly larger than the current study population and if this difference in seroprevalence was observed it would be significant and could lead to a useful increase in test sensitivity and a potential to improve control.
Thank you. We agree with your comment, and we have attempted to suggest this possibility by including changes in the document (present lines 36-38; 213-214). We believe that based on the results obtained, it does not seem to be a very evident effect. However, given that there are significant quantitative differences, it cannot be ruled out that, as we have mentioned before, in certain situations, there may also be qualitative differences.
Lines 208-210: This is a confusing sentence. Does it mean that the board thought it unnecessary to approve the work? Does that mean that the ethical aspects of this study have not been properly considered? What is meant by opportunistic blood and milk samples? Was there another experiment going on at the same time involving these animals?These animals were required to be handled on a total of five occasions to be blood sampled. It is difficult to conceive this intervention as being required for anything other than the experiment. *My concerns are that the need for internal ethical approval was waived because the collecting of samples was “opportunistic”. Ethical review and approval were waived for this study due to opportunistic blood and milk samples taken from animals by veterinary staff and with the farmer´s and owner´s permission. 1. The study required the TB test to be carried out and follow up handling for blood samples on four occasions. It did not state whether the TB test was the scheduled test for statutory disease control, or one specifically carried out for the study. The blood samples were clearly carried out specifically for the study and were not carried out to benefit the 120 animals involved in the study. There is therefore a clear need for ethical approval and for clarification on whether the TB test was carried out specifically for the study. 2. The explanation that the ethical approval was waived because the collecting of samples was “opportunistic” is difficult to understand not least because this is an unusual use of the word opportunistic. In what way was the collection of the samples “opportunistic”. This particular reason may justify approving work, but to waive the need for ethical approval is not logical. 3. If the samples were “opportunistic” there is a clear implication that the animals were being handled for blood sampling for some other reason. This should be explained. If they were not being handled and blood sampled for some other reason these samples were not “opportunistic”.
Like we mentioned previously, we understand and appreciate your concern. We are also committed to animal welfare and conduct all experimental studies in compliance with current legislation, obtaining the necessary permits. In this case, the criteria for considering this study and the animals as experimental were not determined by us but by the responsible ethics committees. In the previously published studies that we have provided, the situation was similar. It appears that the issue arises from the fact that the study was conducted under field conditions, using official tests and reagents administered by official veterinarians involved in eradication programs, and within the framework of an intervention supervised and recommended by official veterinary services to improve the situation of this specific livestock farm. We ensure that the testing intervals are adhered to and are sufficiently spaced out (sampling is done during milking, and the animals are not confined solely for sampling purposes). We believe it is necessary to reiterate that we have extensive experience in requesting animal experimentation permits, and if the corresponding committee had indicated it, we would have applied for it, as we always work in compliance with current regulations. It may indeed be a matter of interpretation, as the reviewer suggests, and their opinion may differ from that of the competent authority. In any case, what the reviewer must understand is that regardless of whether studies require an experimentation permit or not, we always prioritize animal welfare. This also contributes to improving their health status and the quality of the results obtained, whether in experiments or in official control and eradication programs.

Reviewer 2 Report
Comments and Suggestions for Authors
The article is original and very relevant for the field. The authors studied the effect of a comparative intradermal tuberculin test on the humoral diagnosis of paratuberculosis in an infected caprine herd.
The results showed an increase in quantitative response detected in an ELISA technique (booster effect) that were translated into an increase, yet not statistically significant, in the number of reactors. This may improve significantly the diagnostic of tuberculosis and paratuberculosis in field conditions.
The methology of the study is modern and reproducible.
The conclusions are consistent with the evidence and arguments presented.
The references are appropriate, including some relevant authors experience in the field.
I recommend very few minor corrections.
1. Lines 162 and 174 when you cite authors in text, use standard abbreviation (et al.), followed by the number of reference
Author Response
Reviewer 2:
Thank you very much for your comment. We have taken it into account and corrected. Our response to your comment is detailed below.
Lines 162 and 174 when you cite authors in text, use standard abbreviation (et al.), followed by the number of reference.
- Thank you. Cites are corrected (present lines 216, 219 and 227)

Reviewer 3 Report
Comments and Suggestions for Authors
The manuscript entitled “Evaluation of the effect of a recent comparative intradermal tuberculin test on the humoral diagnosis of paratuberculosis using serum and milk samples from goats.” addresses an interesting and difficult topic to address especially due to the different legislation of the various European countries which have eradication plans in place for tuberculosis and therefore have diagnostic difficulties for paratuberculosis.
In fact, it would be appropriate for the Authors to make at least a mention of these legislative aspects in the introductory section.
Furthermore, it would be equally important to refer to the potential heteroallergic and paraallergic reactions to the tuberculin test to outline the picture of possible false positives.
Similarly, a few words should also be said about potential false negatives. All these considerations should be even just mentioned since it is a brief report and not an article in extenso.
Then there are some technical aspects that should be reported:
- Does the ELISA serological test used for the investigation include pre-adsorption with the Mycobacterium phlei antigen to avoid any cross-reactions or not?
-
- Nowhere in the materials and methods or results section is there any mention of the specificity and sensitivity values of the tests. It would be absolutely necessary to have this data.
On the basis of these additions, the Authors will also be able to enrich their discussion, making the work more complete and meaningful.
Author Response
Reviewer 3:
Thank you very much for all your comments and suggestion to improve the quality of the manuscript. Our responses to your questions/comments are detailed below.
The manuscript entitled “Evaluation of the effect of a recent comparative intradermal tuberculin test on the humoral diagnosis of paratuberculosis using serum and milk samples from goats.” addresses an interesting and difficult topic to address especially due to the different legislation of the various European countries which have eradication plans in place for tuberculosis and therefore have diagnostic difficulties for paratuberculosis. In fact, it would be appropriate for the Authors to make at least a mention of these legislative aspects in the introductory section. Furthermore, it would be equally important to refer to the potential heteroallergic and paraallergic reactions to the tuberculin test to outline the picture of possible false positives. Similarly, a few words should also be said about potential false negatives. All these considerations should be even just mentioned since it is a brief report and not an article in extenso.
Thank you. We have included some comments in the introduction and material and methods section regard legislative aspects of PTB and TB in goats (present lines 59-62; 81-87). Although limitations of ELISA sensitivity and specificity (and subsequent false negative and positive reactions, respectively) was mentioned, we have extended this matter (present lines 74-76; 79-80). However, due to the study is mainly focused on PTB diagnosis and the nature of the article (brief report) we cannot elaborate extensively on the CITT limitations. However, if the reviewer considers it necessary to include more information, we can incorporate it in future revisions.
Then there are some technical aspects that should be reported:
- Does the ELISA serological test used for the investigation include pre-adsorption with the Mycobacterium phlei antigen to avoid any cross-reactions or not?
We have included additional information in the manuscript (present lines 149-151)
- Nowhere in the materials and methods or results section is there any mention of the specificity and sensitivity values of the tests. It would be absolutely necessary to have this data.
Thank you. We have included additional information in the manuscript (present lines 154-157)
- On the basis of these additions, the Authors will also be able to enrich their discussion, making the work more complete and meaningful.
Thank you for the suggestion. To enrich and improve our discussion, in the reviewed manuscript we have included new reasons for explaining the different results observed in terms of sensitivity in serum and milk ELISA or even the potencial impact of the lactation stage on the milk ELISA (present lines 230-237). Furthermore, following the comments of the reviewers, we have also interpreted the results of our study taking into account the limitations of our study regarding the sample size and suggesting a potential diagnostic strategy to maximize the detection of PTB-infected animals in other epidemiological settings (present lines 213-214).

Round 2
Reviewer 3 Report
Comments and Suggestions for Authors
I believe the manuscript is ready for publication without further corrections or modifications.